# Development and Validation of an in-line API Quantification Method Using AQbD Principles Based on UV-Vis Spectroscopy to Monitor and Optimise Continuous Hot Melt Extrusion Process

**DOI:** 10.3390/pharmaceutics12020150

**Published:** 2020-02-12

**Authors:** Juan Almeida, Mariana Bezerra, Daniel Markl, Andreas Berghaus, Phil Borman, Walkiria Schlindwein

**Affiliations:** 1Leicester School of Pharmacy, De Montfort University, Leicester LE1 9BH, UK; juan.zelayaalmeida@dmu.ac.uk (J.A.); mariana.bezerra@dmu.ac.uk (M.B.); 2ColVisTec AG, 12489 Berlin, Germany; a.berghaus@colvistec.de; 3Strathclyde Institute of Pharmacy and Biomedical Sciences, University of Strathclyde, Glasgow G1 1RD, UK; daniel.markl@strath.ac.uk; 4EPSRC Future Manufacturing Research Hub in Continuous Manufacturing and Advanced Crystallisation, University of Strathclyde, Technology and Innovation Centre, 99 George Street, Glasgow G1 1RD, UK; 5Product Development and Supply, Medicines Research Centre, GSK, Gunnels Wood Road, Stevenage SG1 2NY, UK; phil.j.borman@gsk.com

**Keywords:** in-line UV-Vis spectroscopy, quality by design, QbD, analytical quality by design, AQbD, hot melt extrusion, HME, analytical target profile development, analytical procedure validation, process analytical technology, PAT, real time release testing, RTRT

## Abstract

A key principle of developing a new medicine is that quality should be built in, with a thorough understanding of the product and the manufacturing process supported by appropriate process controls. Quality by design principles that have been established for the development of drug products/substances can equally be applied to the development of analytical procedures. This paper presents the development and validation of a quantitative method to predict the concentration of piroxicam in Kollidon^®^ VA 64 during hot melt extrusion using analytical quality by design principles. An analytical target profile was established for the piroxicam content and a novel in-line analytical procedure was developed using predictive models based on UV-Vis absorbance spectra collected during hot melt extrusion. Risks that impact the ability of the analytical procedure to measure piroxicam consistently were assessed using failure mode and effect analysis. The critical analytical attributes measured were colour (L* lightness, b* yellow to blue colour parameters—in-process critical quality attributes) that are linked to the ability to measure the API content and transmittance. The method validation was based on the accuracy profile strategy and ICH Q2(R1) validation criteria. The accuracy profile obtained with two validation sets showed that the 95% β-expectation tolerance limits for all piroxicam concentration levels analysed were within the combined trueness and precision acceptance limits set at ±5%. The method robustness was tested by evaluating the effects of screw speed (150–250 rpm) and feed rate (5–9 g/min) on piroxicam content around 15% *w*/*w*. In-line UV-Vis spectroscopy was shown to be a robust and practical PAT tool for monitoring the piroxicam content, a critical quality attribute in a pharmaceutical HME process.

## 1. Introduction

Twin-screw hot melt extrusion (HME) is a continuous process that has been widely used to enhance the solubility of active pharmaceutical ingredients (APIs) [1,2]. This process applies heat and shear energy to produce an amorphous solid dispersion (ASD) of an API into a polymer carrier [3,4]. As a continuous process, HME has the potential to be used in combination with in-line analytical techniques to enable real time monitoring of product quality. The reader is directed to the following reviews of the principles of pharmaceutical extrusion process, as this is not in the scope of the present article [5,6,7].

In-line process analytical technology (PAT) systems are being increasingly used in the pharmaceutical industry since the release of the FDA’s PAT guidance in 2004 for ‘designing, analysing and controlling manufacturing through timely measurements of critical quality and performance attributes of raw and in-process materials and processes with the goal of ensuring final product quality’ [8]. NIR and Raman spectroscopy have been extensively used as PAT tools for the characterisation of in-process materials in HME, including determination of polymer composition in polymer mixtures [9] and quantification of API in a polymer carrier [10,11,12,13]

More recently, UV-Vis spectroscopy that involves light measurements between wavelengths of 200 and 780 nm in transmission or reflection modes has been used as a PAT tool. Transmittance values can be used to calculate a numerical representation of colour by using the International Commission on Illumination (CIE) colour space called CIELAB [14]. The CIELAB colour space represents colour on a three-dimensional orthogonal axis formed by the lightness (L*) and two-colour axis. The colours green to red are described by the axis a*, while blue to yellow are represented by axis b* [14]. Applications of in-line UV-Vis spectroscopy during HME have been reported in the literature for early phase product development [15], feasibility of API quantification [16], residence time distribution determination [17] and monitoring of thermal degradation processes [18]. This technique tends to be both fast to set up and to provide data that is simple to interpret. The short integration time in the millisecond-range delivers rapid results with high sensitivity.

Traditional approaches to analytical procedure validation are centred around a one-off validation exercise in a controlled set of conditions. The quality by design (QbD) methodology has provided a more systematic and risk-based approach for product and process development [19,20]. Currently, this concept has evolved to analytical quality by design (AQbD) that is now being explored by regulators, academia and industry to increase the robustness and promote continuous improvement of analytical procedures [21,22,23]. A recent ICH Q2(R2)/Q14 concept paper proposes the application of AQbD principles to analytical procedure development [24]. This approach emphasizes the importance of predefined method performance requirements prior to commencing analytical development. These requirements are identified in the initial steps of method development and are summarised by the analytical target profile (ATP). Examples of ATPs can be found in a recent publication by Jackson et al. where they proposed a harmonised approach to the use of the ATP concept [25]. The ATP is analogous to the quality target product profile (QTPP) and summarises the performance requirements associated with a measurement (e.g., accuracy and precision) on a quality attribute which needs to be met by an analytical procedure.

Continuous manufacturing platforms, such as HME, are ideal for the implementation of in-line analytical technologies using AQbD. In-line PAT can be embedded in quality systems to provide monitoring of quality and to enable the development of control strategies for real time release testing (RTRT). This should be based on an in-depth understanding of the relationship between process parameters, in-process material attributes and product attributes, as described in the European Medicines Agency guideline for RTRT [26].

The ICH Q2(R1) defines the methodology for validation of analytical procedures, mostly based on HPLC [27,28], but it does not provide specific guidance on PAT methods. The revision to ICH Q2 is intended to include validation of spectroscopy techniques reliant on multivariate models [24]. Alternative validation approaches for spectroscopic techniques have been used and referred to as the ‘accuracy profile’. This validation approach was developed by the Societé Française des Sciences et Techniques Pharmaceutiques (SFSTP) based on trueness and precision. The accuracy profile has been successfully applied to analytical procedure validation of on-line and at-line NIR and in-line Raman in HME processes [29,30,31,32,33].

Previous work published by our group reported the benefits of using in-line UV-Vis spectroscopy as a rapid analytical technology with applications in the early phase product development of HME processes [15]. Following this work, optimisation of the manufacturing process of piroxicam (PRX) in Kollidon^®^ VA 64 (KOL) was carried out for the intended QTPP of immediate release tablets containing 20 mg of API. The design space for the identified critical process parameters (CPPs) and materials attributes was stablished as: concentration of API in the polymer carrier: 10 ≤ API ≤ 20% *w*/*w*; barrel and die-temperature range 130–140 °C; API/polymer mixture feed rate 5–10 g/min; and screw speed 200–300 rpm. Based on this information, the process conditions selected to develop the analytical procedure were barrel temperature profile 120–140 °C, die-temperature 140 °C, API/polymer mixture feed rate 7 g/min and screw speed 200 rpm.

In this paper, AQbD principles and an accuracy profile approach are applied to develop and validate a method to quantify piroxicam content in a Kollidon^®^ VA64 carrier using in-line UV-Vis spectroscopy during HME. This includes a failure mode effect analysis of the analytical procedure and the development of an ATP to determine the content of piroxicam in Kollidon^®^ VA 64.

## 2. Materials and Methods

### 2.1. Materials

Piroxicam (Medex, Rugby, UK) was the active ingredient and Kollidon^®^ VA64 (donated by BASF, Ludwigshafen, Germany) the polymer carrier. Stock mixtures of 32% *w*/*w* PRX in KOL were prepared. These were later diluted by adding further KOL to reach the desired concentrations. The powder mixtures were thoroughly blended using a V-cone mixer (Pharmatech, Coleshill, UK) for 10 min and using 75% of the volume fill. Off-line UV-Vis calibration curve was performed to check content uniformity and 10 min was considered the optimal time for blending. The rationale for using piroxicam and Kollidon^®^ VA64 is described in [15].

### 2.2. Extrusion Setup

The extruder used in the HME process was a Leistritz Nano 16 (Somerville, NJ, USA), which is a co-rotating twin screw extruder (screw diameter 16 mm) with three heating zones and a die zone. The feeder used was an FW20 FlexWall feeder (Brabender Technologie, Duisburg, Germany). A schematic diagram of the hot melt extrusion process is presented in Figure 1. Optimised process conditions (extruder temperature 120 (zone 1), 130 (zone 2) and 140 °C (zone 3 and die), screw speed 200 rpm and feed rate 7 g/min) were used to develop and validate the analytical procedure for API around 15% *w*/*w*. The method robustness was tested by evaluating the effects of screw speed (150–250 rpm) and feed rate (5–9 g/min) on piroxicam content around 15% *w*/*w*.

### 2.3. In-line UV-Vis Spectroscopy

The UV-Vis spectrophotometer (Inspectro X ColVisTec, Berlin, Germany) was setup using the optical fibre cables with two probes (TPMP, ColVisTec, Berlin, Germany) installed into the extruder die in a transmission configuration, as presented in Figure 1. A reference UV-Vis transmittance signal was obtained with empty die at the selected process temperature, 140 °C. Transmittance data was collected from 230 to 816 nm with a resolution of 1 nm. Data collection frequency was 0.5 Hz and each spectrum was taken as the average of 10 scans. The spot size of the used UV-Vis spectrophotometer probes was 2 mm diameter and measured sample volume of typically 2.5 mm^3^.

As described earlier, CIELAB colour space is an approach to express colour using three parameters: ‘lightness’ (L*), ‘green to red’ (a*) and ‘blue to yellow’ (b*) defined by the International Commission on Illumination (CIE). These parameters are calculated from the UV-Vis transmittance spectra in the range from 380 to 780 nm. Figure 2 illustrates the spectral tristimulus values $\bar{X}$, $\bar{Y}$ and $\bar{Z}$ that are used to calculate X, Y and Z using Equations (1)–(4), where T is the transmittance spectrum, S is the relative spectral power and λ is the wavelength. Then, L*, a* and b* are calculated using Equations (5)–(7), where X_n_, Y_n_ and Z_n_ are the spectral tristimulus values of the nominally white object. The values of $\bar{X}$, $\bar{Y}$, $\bar{Z}$, S and further explanation are available in [14].
(1)$$X=k\sum_{\lambda }T\left(\lambda \right)S\left(\lambda \right)\bar{X}\left(\lambda \right)\Delta \lambda ,$$
(2)$$Y=k\sum_{\lambda }T\left(\lambda \right)S\left(\lambda \right)\bar{Y}\left(\lambda \right)\Delta \lambda ,$$
(3)$$Z=k\sum_{\lambda }T\left(\lambda \right)S\left(\lambda \right)\bar{Z}\left(\lambda \right)\Delta \lambda ,$$
(4)$$k=100/\sum_{\lambda }S\left(\lambda \right)\bar{Y}\left(\lambda \right)\Delta \lambda ,$$
(5)$$L^{*}=116\times \sqrt[{3}]{\frac{Y}{Y_{n}}}-16,$$
(6)$$a^{*}=500\times \left(\sqrt[{3}]{\frac{X}{X_{n}}}-\sqrt[{3}]{\frac{Y}{Y_{n}}}\right),$$
(7)$$b^{*}=200\times \left(\sqrt[{3}]{\frac{Y}{Y_{n}}}-\sqrt[{3}]{\frac{Z}{Z_{n}}}\right).$$


### 2.4. HPLC Reference Analytical Procedure

HPLC was performed as an off-line method to assay PRX content in the extruded samples. Assay values obtained from this reference method were used as true content values in the development and validation steps of the UV-Vis quantification method. The equipment used was an Agilent 1100 system with a Phenomenex C-18 column (Kinetix 4.6 × 250 mm, 5 μm); and methanol:sodium dihydrogen phosphate buffer 45:55 mobile phase (LiChropur, Millipore, Waltford, UK); pH 3.0; flow rate 1.2 mL/min. A calibration curve was constructed using PRX standard solutions with concentrations ranging from 10 to 80 μg/mL and the measured respective peak areas (mAU*s) at 360 nm. The milled extrudate (312 mg) was dissolved in 100 mL of 0.01M methanolic hydrochloric acid (Fisher, Loughborough, UK), which was diluted by a factor of 10 to obtain a sample solution. Sample solutions were homogenised by mechanical agitation (Stuart, Stone, UK) for 5 min. All samples were prepared in triplicate.

### 2.5. UV-Vis Spectra Data Analysis Tools

Analysis of the in-line UV-Vis spectral data was performed using Matlab (Matlab R2018a, Natick, MA, USA) functions for multivariate analysis with mean-centred data, for method validation criteria calculation and data plotting.

### 2.6. Experimental Design

The experimental design for the method development combined one calibration and two validation data sets that were collected on different days. The calibration experiment comprised of five defined PRX concentrations of 10.58, 12.46, 14.45, 16.54 and 18.46% *w*/*w*. The validation data sets comprised of four PRX concentrations within the calibration data set range with concentrations of 11.66, 13.45, 15.45 and 17.50% *w*/*w* produced on different days. All PRX-containing samples were analysed by off-line HPLC assay and reported as true values for the used concentration ranges. It is important to note that the API concentration range is not restricted to high values. Feasibility studies performed by our research group show that this PAT tool can also detect low analyte concentrations, but this is out of the scope of this paper.

The robustness of the proposed method was evaluated by performing experiments using samples with 14.5% *w*/*w* of PRX and varying feed rate (5–9 g/min) and extrusion screw speed (150–250 rpm) within the design space range obtained from the process optimisation, as described at the end of the introduction section. The critical analytical attribute; parameter b*, was used to determine steady state during HME process and absorbance values were used to build the calibration model described in the next section.

## 3. Results and Discussion

### 3.1. Analytical Quality by Design

A quality by design approach to analytical method design was used to develop and validate an analytical procedure. This included defining the method performance requirements via an ATP [34] and the use of structured, risk-based approach to method development and evaluation using failure mode and effects analysis and multivariate experimental design.

#### 3.1.1. Analytical Target Profile

The ATP concept considers the maximum permitted measurement uncertainty associated with the reportable value, based on the accuracy and precision performance requirements of the analytical method. The analytical target profile (ATP) was to measure the content of piroxicam in Kollidon^®^ VA64 carrier during HME process as showed in Table 1. The limits presented in the ATP show a combined accuracy and precision uncertainty of ±5% with 95% probability. Also, these accuracy and precision limits are commonly accepted by the pharmaceutical regulatory agencies [25].

#### 3.1.2. Failure Mode and Effects Analysis of the Analytical Procedure

The goal of this risk analysis was to understand and control potential sources of variation, so the robustness of the analytical procedure can be evaluated, improved and verified. The scores for criticality were based on severity (S), occurrence (O) and detectability (D). A risk priority number (RPN) was calculated by multiplying the three scores, i.e., RPN = S × O × D. The RPN is used to identify the most impactful failure modes through ranking. It should be noted that the composition of the team performing the Failure Mode Effect Analysis (FMEA) can influence the rankings, however, by using the same team and ranking method before and after the implementation of the controls, the RPN scores before and after the implementation of controls/mitigations will be subjected to the same scoring decisions.

The severity scale was based on the impact that the sources of variability (identified previously through using an Ishikawa Diagram [21]) have on the ability of the analytical procedure to measure the PRX content (i.e., impact the ability to meet any of the criteria in the ATP, see Table 1.

The scales for ranking were: Severity: 1 = not severe, 4 = slightly severe, 7 = moderately severe, 10 = extremely severe; occurrence: 1 = infrequently, 4 = fairly infrequent, 7 = fairly frequently, 10 = frequently; detectability: 1 = almost certain, 4 = highly likely, 7 = moderate, 10 = impossible. Severity scores of 7 or 10 represent failure modes where a large or small change respectively in the variable has a significant impact on some of the ATP criteria.

The main focus of risk mitigations activities was to reduce the occurrence and/or improve the detection of these failure modes. The results before and after mitigations were implemented are presented in Table 2. For example, for the probes/fibres set up (Table 2, index number 1) precision o-ring spacers have been used to provide the same gap between the probes. The changes of gap size as a function of temperature and cleanness of the probe lenses have also been investigated. The probe setup procedure was part of the standard operational procedure (SOP) for the manufacturing process. Any variation in gap size, e.g., from different day setup was within the analytical method validation model. The average gap size for the experimental conditions used was 0.77 ± 0.05 mm.

Seven potential failure modes were initially identified as high risk (RPN ≥ 196). Standard operations following good laboratory practice and specific data analysis procedures that were identified must be followed to mitigate the risks related to probe temperature, steady-state determination, sample selection within steady state and variability between different days samples. Spectral variable selection is equally important and only the parts of the spectra that describe changes in API content should be selected using the loadings of the first principal component (PC1). Under or over-fitting of partial least squares regression (PLS) model can also be critical to the accuracy and precision of the model. This can be mitigated by holdout cross-validation with 20% of the data set used to test and optimise the number of latent variables used in the PLS model.

After risk mitigation, all seven risk areas identified as high risk could be controlled to acceptable risk level scores (RPN ≤ 40). Figure 3 shows the RPN scores before and after the implementation of control measures for the method.

### 3.2. Calibration Model Development

The procedure to build predictive models to read out real-time API concentration during HME process using in-line UV-Vis spectra is summarised in Figure 4. The methodology consists of four main data processing steps, which are sample selection, normalisation, variable selection and number of latent variables. Details of each step are presented in the sub-sections below.

For this UV-Vis analytical procedure, the samples are the spectra collected during the HME process and variables are the spectral wavelengths of interest. A normalisation procedure was developed to improve method intermediate precision. Also, a methodology to prevent over and under-fitting in partial least squares models was applied.

#### 3.2.1. Sample Selection

Steady state detection is difficult to achieve during continuous processing, and it is usually demonstrated through time defined end points [8]. When the data are acquired during the transition period, the predictive model trueness and precision are negatively affected by the unstable signal. For the data sample selection, it is important to remove measurements that did not reach steady state in both calibration and validations data sets. Steady-state condition during the HME process was determined using b* colour parameter from UV-Vis data. A linear positive correlation between PRX content and b* values is shown in Figure 5. The data in the visible region (380 to 780 nm) of the spectrum is used to calculate the b*, and as presented in Figure 6 is where the change because of the amount of PRX occurs. However, if the API does not absorb in the visible region, a principal component analysis (PCA) can be used to detect the variations in the spectra after data acquisition and improve data set precision and accuracy. There are noticeable differences in the absorbance that enable quantification of the PRX across the range studied.

Principal component 1 (PC1) of the absorbance spectra represents the greatest variations within the data set [35]. As the only variable investigated is the API content, the PC1 scores are correlated to the variations of the absorbance spectra caused by this parameter (Figure 7). Figure 8 shows the PC1 scores as a function of acquisition time from low to high concentrations of PRX with data in transition period. The grey points show the transition from pure KOL used to clean the extruder between runs to the next PRX sample. Figure 9 depicts PC1 scores only in the steady-state region and the explained variance of the first principal component is improved from 92.88% (Figure 8) to 93.52% (Figure 9). This means that the PLS model will be enhanced because less unwanted spectral variability will be added to it. Hence, when PC1 scores reach a plateau, this means that the ASD has a stable API content and the process is in steady state. By using the PC1 scores instead of time defined end points, the method become more accurate and precise.

#### 3.2.2. Normalisation

The reference UV-Vis spectrum was obtained in a cleaned, empty extruder, producing a baseline for the transmittance measurements, however when the equipment is calibrated on a different day, this baseline may slightly change. To minimise day to day variability, KOL readings from each experiment day were used to normalise the transmittance spectra of the samples. Studies with Raman and NIR use other common pre-processing techniques such as derivatives [30], standard normal variate transformation [32] and multiple scatter correction [36]. These techniques can be complex to implement [37] and for the UV-Vis were not required. Only normalisation was used to pre-process the spectra.

Polymer batch variation should not influence the results as long as the same batch is used for the reference and API/polymer mixtures. The normalisation procedure using the polymer spectra should also eliminate effects of inter-batch polymer variations as only the normalised spectrum is used to determine the API content.

After the normalisation, the transmittance, T, collected from the UV-Vis spectrometer was converted to absorbance, A, using the Beer–Lambert’s law:
(8)$$A=-\log_{10}\left(T\right).$$


#### 3.2.3. Spectral Variable Selection

Spectral variable selection is performed in spectroscopic techniques, because otherwise the model would be affected by unwanted variability in the spectra instead of the analyte of interest [38]. Therefore, quantitative models based on Raman and NIR uses selected spectrum ranges [30,31,32,33], hence the same is performed for UV-Vis in this study.

The wavelength range was selected to maximise the contribution of the API to the absorbance spectrum features and minimise the effects of unrelated information. There are many methods for variable selection [39,40] and PCA was applied in this work because of its simplicity.

A PCA was applied to the calibration set and the PC1 loadings were used to select the wavelength range with high signal to noise ratio, because in a univariate experiment the PC1 loadings indicated that the variables were mainly influenced by the API content.

Figure 10a present the loadings of PC1 using the entire spectrum for the PCA. It is noticed that the loadings are close to 0 below 446 nm and above 540 nm. Using only the wavelength range from 446 nm to 540 nm for the PCA, the percentage of variation explained by PC1 is improved to 98.38% compared to 93.52% when using the entire spectrum (Figure 10b). The use of selected variables assisted the conformance of the method with the ATP.

#### 3.2.4. Number of Latent Variables

The number of latent variables (LVs) used in a PLS calibration model is important, as too many or too few may cause over or under-fitting of the model. The number of LVs was selected based on the model population analysis developed by Deng et al. [41].

The calibration data set was randomly divided into two sets, one set containing 20% of the data that was assigned to a cross-validation subset and the remaining 80% to a calibration subset. Then, ten PLS models were built using the calibration subset with 1 to 10 LVs. The coefficient of determination (R_cv_^2^) and root mean square error (RMSECV) of the cross-validation subset were calculated for each PLS model, in a similar fashion as reported in literature for other spectroscopy techniques [30,31,32].

The R_cv_^2^, RMSECV and regression vector were plotted versus the number of LV used in the model as presented in Figure 11. The number of LV with maximum R_cv_^2^ and minimum RMSECV was computed using the data presented in Figure 11a,b, located at the end of the elbow of the charts. Additionally, visual selection of the regression vector with less incorporated noise (Figure 11c) supported the previous observation. Hence, four LVs were selected to use in the PLS model.

#### 3.2.5. Predictive Model

A PLS calibration model was developed using true API content validated with HPLC as responses (Y) containing 10.58, 12.46, 14.45, 16.54 and 18.46% *w*/*w* of PRX in KOL. The predictors (X) of the PLS model were 600 absorbance spectra, 120 for each concentration level, in the range of 446 to 540 nm and 4 LV. The calibration model R_c_^2^ and RMSEC were respectively 0.9990 and 0.0905% *w*/*w*.

The developed methodology to build PLS predictive models using in-line UV-Vis spectra based on PCA and cross-validation linearity have previously been applied to Raman and NIR spectra as reported in [30,31,32,33,35,42]. The methodology to build predictive models using in-line UV-Vis spectra can be widely applied to other APIs and polymers during production of ASD using HME. This methodology can offer analytical scientists a useful case study on how to tackle the validation of similar methods deploying the use of PLS.

### 3.3. Analytical Procedure Validation

The quantitative analytical procedure validation used here was based on ICH Q2(R1) guideline and the accuracy profile concept suggested by an SFSTP commission as reported in the literature [32,33]. This commission produced a summary report on the validation of quantitative procedure divided in three parts. In the first part, the authors proposed the accuracy profile, as a strategy to harmonize approaches to analytical procedure validation [43]. In the second part, the commission presents a protocol for experimental design required for the method validation based on the accuracy profile [44]. In the third publication, the commission provides a numerical example that illustrates all the steps to build an accuracy profile [45].

The validation sets comprised four PRX concentration levels with 50 UV-Vis replicate measurements for each concentration, on each day. The true PRX concentration, measured by HPLC, and the mean predicted concentration from the UV-Vis model for each validation day are in close agreement as presented in Table 3. The UV-Vis based method and the standard analytical technique produced comparable results. For the validation criteria and accuracy profile the true and predicted PRX contents used were averages of the results obtained in both days.

#### 3.3.1. Validation Criteria

The parameters defined in ICH Q2(R1) for the validation of an analytical procedure for quantification of API are trueness, precision, linearity, limit of quantitation and range [46]. Furthermore, uncertainty and total error were calculated. Robustness was also assessed as discussed in Section 3.3.3.

The trueness represents the closeness of the average to the true value [47]. The precision of an analytical procedure is the closeness of agreement amidst same sample measurements from various sampling under the same process conditions. Linearity is the ability of an analytical procedure to produce results that are directly proportional to API content in the required range. The limit of quantification is the lowest sample concentration that can be quantitatively determined with appropriate accuracy and precision [46].

The uncertainty is a dispersion of measured values from the expected value. It is calculated as uncertainty of the bias, measurement uncertainty and expanded uncertainty [47]. The total error is the sum of the absolute bias and intermediate precision standard deviation, which are the effects that cause the measured value to be different from the true value.

The validation criteria were calculated using the true and predicted PRX contents average of the results obtained in both experiment days (Table 3). The results for trueness presented as relative bias and recovery are displayed in Table 4. Relative bias varied from −0.29 to 0.47%, markedly lower than the limit ±2% defined in the ATP. The recovery results ranged from 99.71 to 100.47% from the true value.

The precision is related to the ruggedness of the method [48] and is obtained with repeatability and inter-day intermediate precision [49], both calculated through relative standard deviation (RSD). The highest value for repeatability and intermediate precision were 0.80% and 1.09% lower than 1.8% defined in the ATP (Table 4). The RSD% values of both intermediate precision and repeatability were close, demonstrating that day-to-day variability did not contribute much noise to the spectral data [30]. These results demonstrate that the mitigation strategies presented in Table 2 prevented failure modes to impact the method precision. The relative β-expectation tolerance limits are calculated based on the trueness and precision; these are used to build the accuracy profile presented in the next section.

Linearity is presented in Figure 12 for the introduced concentration and the predicted concentration by the model. The results obtained for R_v_^2^ and RMSEV are 0.9963 and 0.1108% *w*/*w*, showing that the spectral correlation with API content is preserved across data sets produced on different days.

The total error is presented as absolute and relative total error and the results are tabulated in Table 5. The values of absolute and relative total error did not exceed 0.22 and 1.42%, respectively.

The results for uncertainty are obtained with uncertainty of the bias, measurement uncertainty and expanded uncertainty. Table 5 shows all uncertainty results and illustrates that the dispersion of the measured values was very low. The limit of quantitation was 11.66% *w*/*w* of PRX, since this was the lowest validated API content.

The trueness and precision of the UV-Vis are similar to the NIR method presented in [30], furthermore Raman results for these two criteria are slightly worse than the values presented here [31,32,33]. The R_v_ and RMSEV obtained in this study were better than the results reported using NIR and Raman spectroscopy [30,31,32,33].

#### 3.3.2. Accuracy Profile

The accuracy profile is a visual tool to illustrate the analytical procedure performance. This approach applies total error to evaluate the risk of measurement failure. The β-expectation tolerance limits are calculated for each concentration level and 95% of the measurements should fall within the acceptance limits [50]. The acceptance limit, λ (Equation (9)), is the difference between the predicted API concentration by the model $\left(y_{i}^{\prime }\right)$ and the true value $\left(y_{i,t}\right)$, and was set to 5% for API determination [43,47,51].
(9)$$\left|y_{i}^{\prime }-y_{i,t}\right|<\lambda$$


The values of relative bias of every measurement and mean relative bias for each concentration level are plotted along with the acceptance limits and β-expectation tolerance limits to build the accuracy profile [47,52].

The 95% β-expectation tolerance limits obtained in the accuracy profile for PRX content is within the acceptance limits of ±5% for determination of active ingredients in dosage forms as observed in Figure 13 and Table 4. This result indicates that the mitigation strategies presented in Table 2 were adequate to develop an analytical method with enough ruggedness and robustness to comply with the ATP presented in Table 1.

The UV-Vis method β-expectation tolerance limits were within ±5% acceptance limits. Other authors have applied accuracy profile for quantitative method validation using inline spectroscopy techniques during HME, however the acceptance limits used by them was ±10% [32] and ±15% [33]. These studies were performed using Raman and high drug loads.

#### 3.3.3. Robustness

The robustness of a method is its ability to remain unaltered by deliberate variations to the optimised parameters [46]. As mentioned at the end of the introduction section, a previous study was done to define the optimised conditions of the HME process using PRX in KOL. A design of experiment was performed to assess the robustness of the in-line UV-Vis analytical procedure under the process conditions within the design space for optimised conditions.

Table 6 presents the predicted PRX content, feed rate and screw speed from the robustness experiment. The concentration limits were defined based on the maximum mean relative bias value defined in the ATP, that was ±2%. Hence, the lower and upper limits were determined as 14.15 and 14.75% *w*/*w*, respectively. The mean predicted API concentration relative bias in the robustness design of experiments (DoE) exceeded the limit defined on the ATP in runs 1, 3, 4, 7, and 8.

A contour profiler is a graphical tool to display the influence of process parameters on a response variable. Here, the profiler was used to calculate the impact of feed rate and screw speed upon the concentration of PRX predicted by the PLS model (Figure 14). The white area is where the predicted content is within the relative bias acceptable limits defined by the ATP. According to the contour profiler, it is possible to work within a broad feed rate range but narrow screw speed range. Therefore, the screw speed has a higher impact than feed rate on the predicted result.

A method operable design region (MODR) is the region in which parameter variations do not affect the performance of the method [34]. The contour profiler was used to support the determination of the MODR. To produce samples within the ATP, the MODR for screw speed and feed rate were defined in the green area of Figure 14. The feed rate and screw speed process variables impact the method, but they have been constrained across an optimised range. The corners of the MODR are presented in Table 7, with the lower and upper screw speed limit linearly increasing with the increase of feed rate.

The robustness DoE was deliberately defined over a wider range (where the method performance meets the ATP [34]) to gain further knowledge of the conditions and greater operational flexibility. The MODR was defined using narrower ranges (Table 7) than used in the DoE. The MODR identified in the robustness experiments included ranges of screw speed and feed-rate that spectral data met the relative bias limit established in the ATP (Table 1). Therefore, the in-line UV-Vis analytical method was successfully developed and validated for in-line monitoring of PRX content in HME process underpinned by the AQbD approach.

## 4. Conclusions

An in-line UV-Vis method for API quantification during hot-melt extrusion processes was developed and validated according to the AQbD framework. AQbD concepts have not typically been applied to the PAT methods and the use of the analytical target profile to define and validate the performance requirements of an in-line method is novel. Failure mode and effects analysis was used to mitigated the risks during analytical procedure development and validation. Additionally, the AQbD approach is aligned with latest industry and regulatory thinking.

The predictive model was developed using a multivariate approach and an innovative methodology that assisted the analytical procedure agreement with ICH Q2(R1) criteria and accuracy profile validation strategy.

The results show that UV-Vis spectroscopy for HME is less complex than techniques such as NIR and can reliably detect variations in piroxicam content in a pharmaceutical HME product. The same methodology for the development and validation can be applied to different API/polymer systems to develop quantitative models based on UV-Vis for in-line HME monitoring.

Polymer batch variation should not influence the results as long as the same batch is used for the reference and API/polymer mixtures. Risk mitigation strategies can prevent failure modes such as probe gap size, probe temperature and polymer batch variations that can impact the method precision. The feed rate and screw speed process variables impact the method, but they have been constrained across an optimised range.

This in-line PAT can be used for real time monitoring of API content during HME continuous manufacturing process. The accuracy profile obtained with two validation sets (from different days) showed that 95% of the future measurements will fall within ±5% acceptance limits.

The stepwise approach from defining the performance requirements, optimising the method and validating the predictive model offers analytical scientists a useful case study on how to tackle the validation of similar methods deploying the use of PLS.

## Figures and Tables

**Figure 1 pharmaceutics-12-00150-f001:**
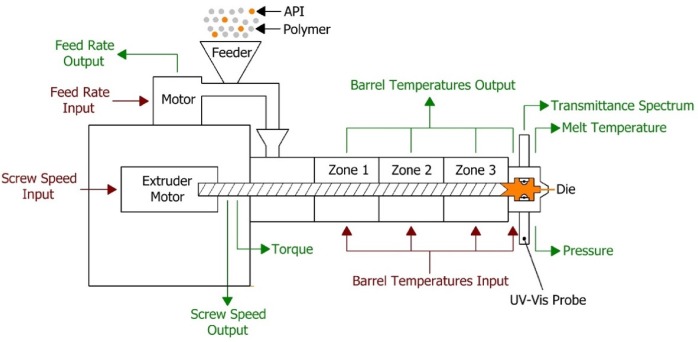
Schematic of the hot melt extrusion process.

**Figure 2 pharmaceutics-12-00150-f002:**
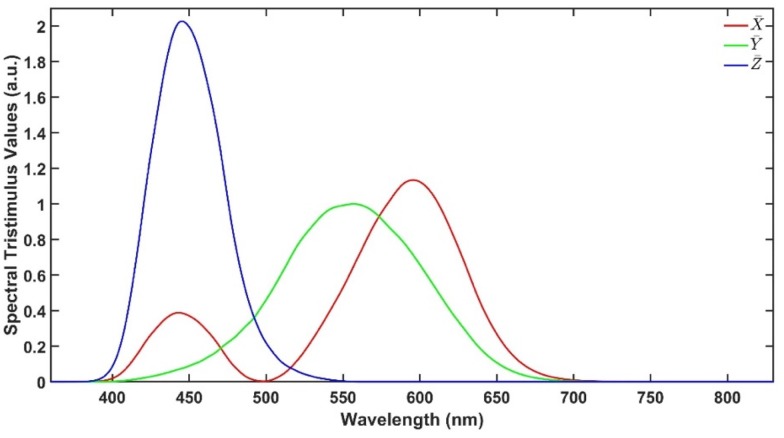
Spectral tristimulus values.

**Figure 3 pharmaceutics-12-00150-f003:**
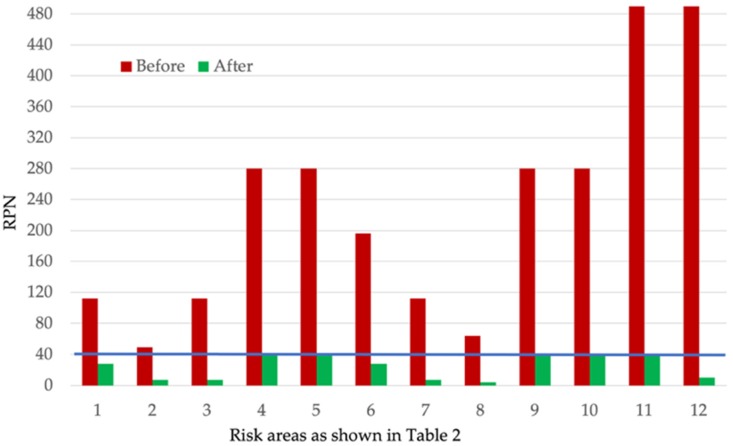
Risk priority number (RPN) values of the risk areas before and after control strategy implementation as described in Table 2 for the method.

**Figure 4 pharmaceutics-12-00150-f004:**
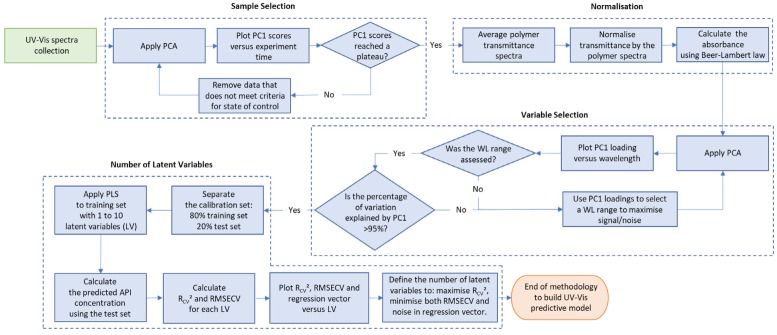
Methodology to build predictive model using UV-Vis spectra. WL= wavelength, PCA = principal component analysis, R_cv_^2^ = coefficient of determination, RMSECV = root mean square error, CV = cross-validation, LV= latent variables.

**Figure 5 pharmaceutics-12-00150-f005:**
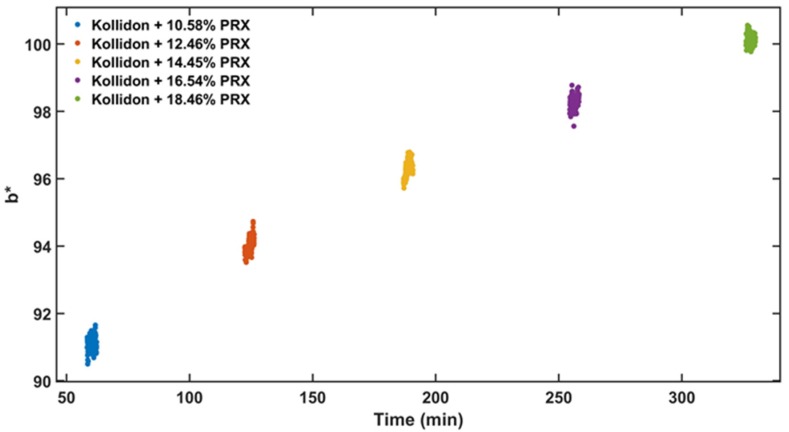
b* (blue to yellow colour parameter) vs. experiment time of the calibration data set.

**Figure 6 pharmaceutics-12-00150-f006:**
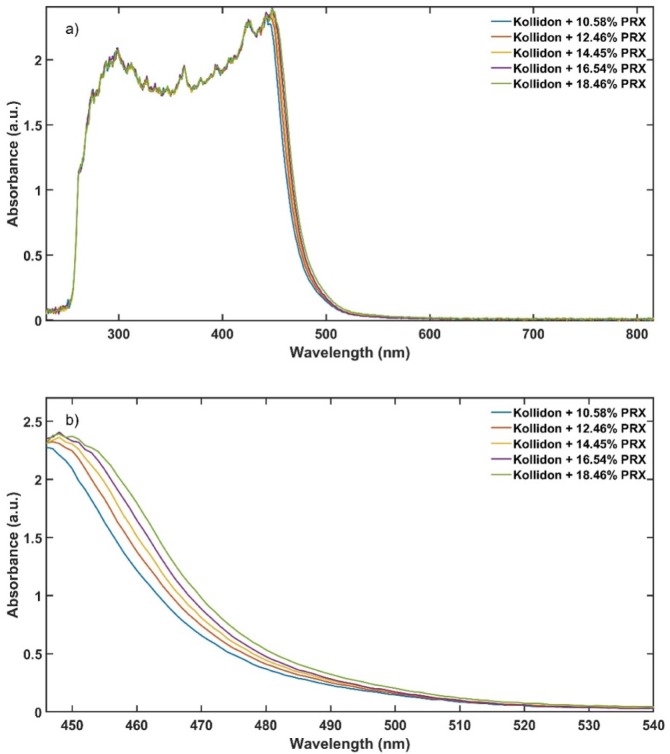
Absorbance spectra of the calibration data set, (**a**) full spectrum and (**b**) 446 to 540 nm range.

**Figure 7 pharmaceutics-12-00150-f007:**
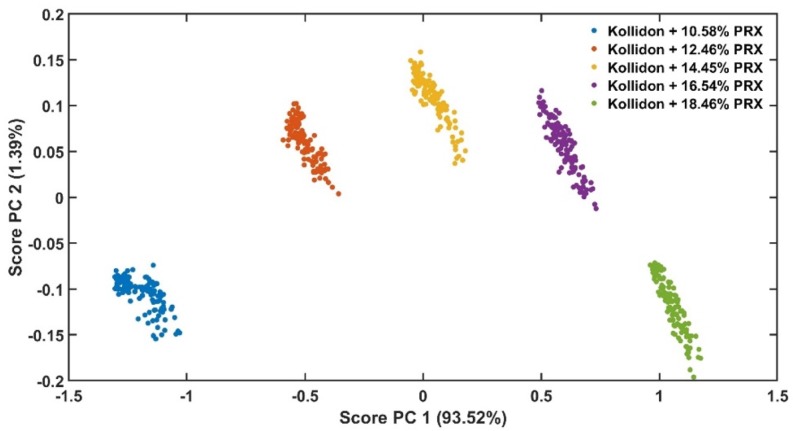
Scores of principal components (PC) 1 vs. PC 2 of the calibration data set.

**Figure 8 pharmaceutics-12-00150-f008:**
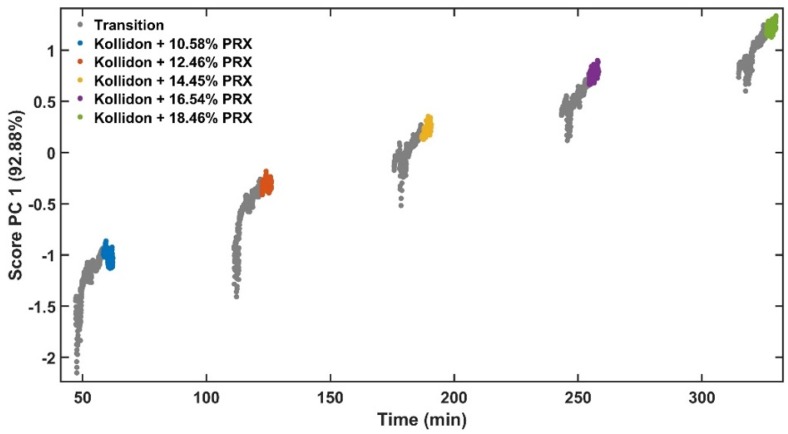
Scores of PC1 vs. experiment time with transition data.

**Figure 9 pharmaceutics-12-00150-f009:**
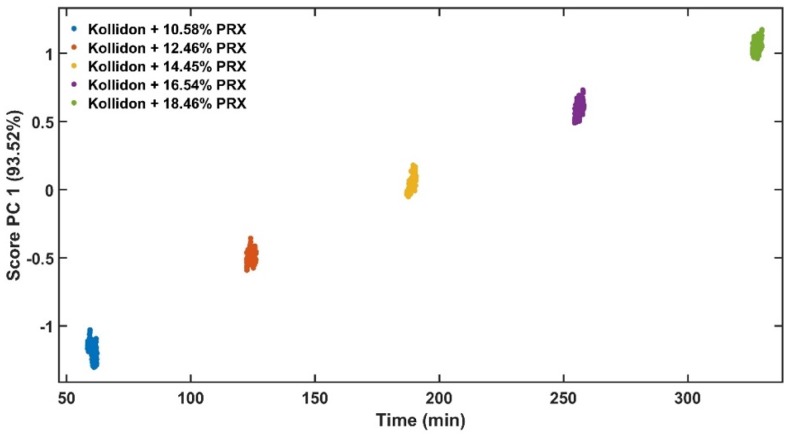
Score of PC1 vs. experiment time only in steady state.

**Figure 10 pharmaceutics-12-00150-f010:**
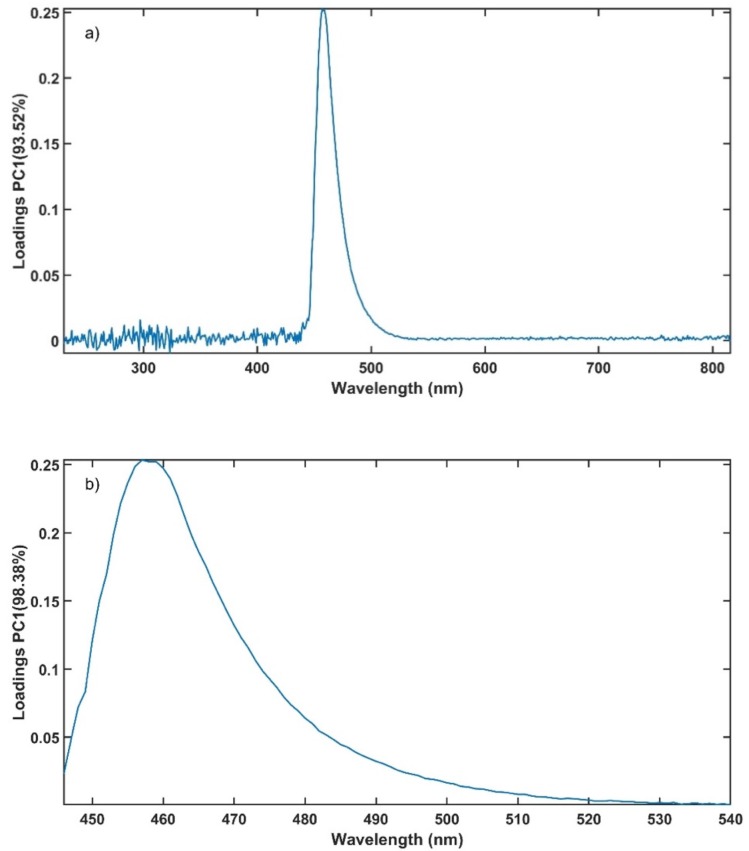
Loadings from a PCA of the calibration data set using (**a**) the entire wavelength range and (**b**) the selected range.

**Figure 11 pharmaceutics-12-00150-f011:**
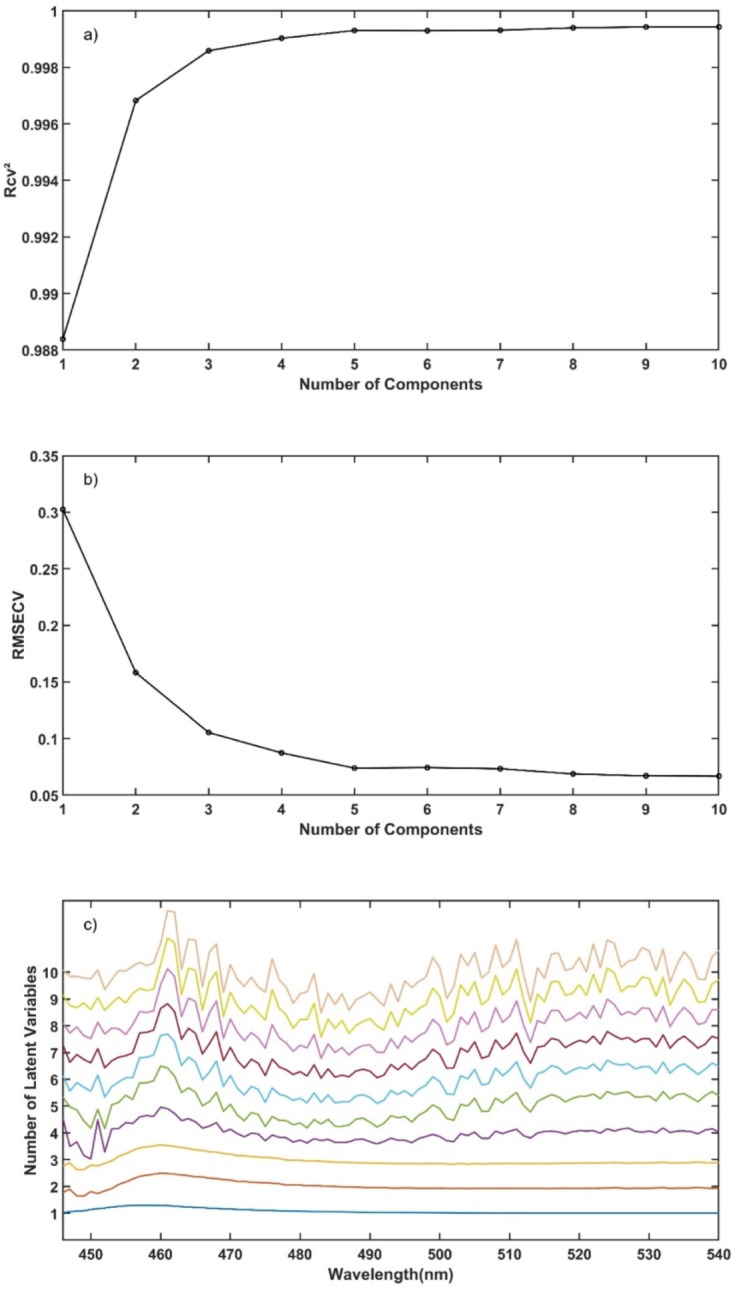
(**a**) R_cv_^2^ vs. number of components, (**b**) RMSECV vs. number of components and (**c**) stacked PLS regression vector for different number of latent variables vs. wavelength.

**Figure 12 pharmaceutics-12-00150-f012:**
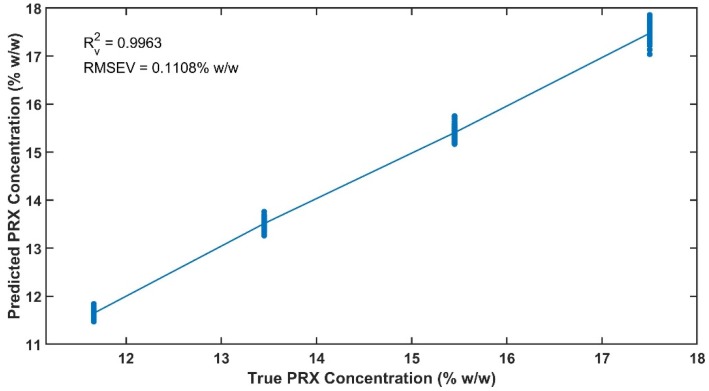
Linearity of the validation data set.

**Figure 13 pharmaceutics-12-00150-f013:**
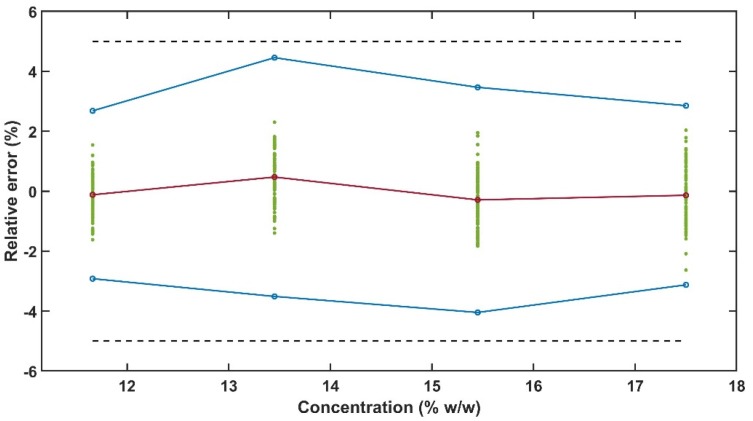
Accuracy profile with the dashed black lines indicating ±5% limits, the blue lines with circles are the β-expectation tolerance limits, green points are the relative bias for every measurement and the red line with circles is the method mean relative bias.

**Figure 14 pharmaceutics-12-00150-f014:**
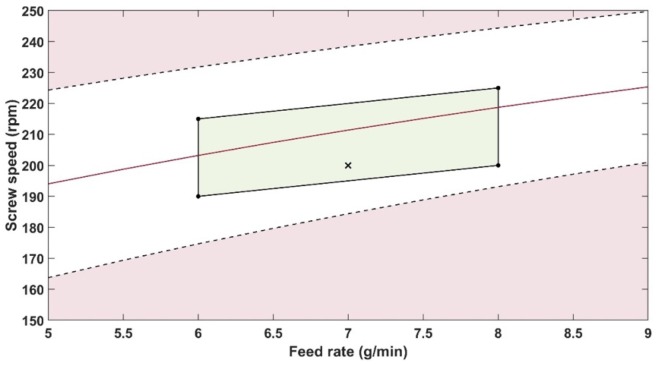
Contour profiler showing the influence of the interaction of screw speed and feed rate on the predicted PRX amount using the PLS model. The red line indicates the predicted PRX content of 14.45% *w*/*w*. The black cross marks the optimised extruder parameters. The black dashed lines indicate the upper and lower limits defined respectively 14.75 and 14.15% *w*/*w* of PRX.

**Table 1 pharmaceutics-12-00150-t001:** Analytical target profile (ATP) to determine the content of piroxicam (PRX) in Kollidon^®^ VA 64 (KOL).

Attribute Range Requirements (Criteria)	Attribute Range Requirements ^1^ (Rationale)	Specificity (Criteria)	Accuracy Requirement (Criteria)	Accuracy Requirement (Rationale)	Precision Requirement (Criteria)	Precision Requirement (Rationale)
Content 80–120% label claim (LC)	Covers typical content specification range of 95.0–105.0% LC	Specific to Piroxicam in the presence of Kollidon^®^ VA64	Mean relative bias of ≤2.0% LC of theoretical across the attribute range	Ensures difference between true and estimated mean is within the specification range and allows adequate proportion of widest specification range for analytical and process variability	Relative standard deviation (RSD) ≤1.8% across the attribute range	Ensures that the analytical variation around the estimated mean lies within the widest specification range

^1^ Attribute range is typically 90.0–110% for US market and 95.0–105.0% for EU.

**Table 2 pharmaceutics-12-00150-t002:** FMEA for the analytical procedure. The ATP performance characteristic affected are specificity, accuracy and precision.

Index Number	Risk Area	Potential Failure Mode	Potential Failure Effects	S	O	D	RPN	Mitigations	Revised Ranking			
									S	O	D	RPN
1	Probes/Fibres	Probe position (gap) and cleanliness. Fibres alignment and movement.	Accuracy and precision	7	4	4	112	Measure gap size with feeler gauge. Clean optical lenses after this process. Fix fibres to avoid movement. Make sure the probes are aligned and the optical lenses are clean.	7	4	1	28
2	UV-Vis spectrometer	Number of scans averaged and data collection frequency. Noise level	Accuracy and precision	7	1	7	49	Optimise values for the process.	7	1	1	7
3	UV-Vis spectrometer	Number of lamp flashes. Saturation of light	Accuracy and precision	7	4	4	112	Optimise values for the process. Follow guideline from equipment supplier.	7	1	1	7
4	UV-Vis spectrometer	Variable blank measurement for different day experiments	Accuracy and precision	10	7	4	280	Take measurement following standard operating procedures.	10	4	1	40
5	UV-Vis spectrometer	Probe temperature changes causing variability on the reference spectrum	Accuracy and precision	10	7	4	280	Wait for the signal to stabilise. Make sure the die temperature is stable. Perform the blank reference again, if necessary.	10	4	1	40
6	Data management	Steady-state determination. Signal to noise ratio	Accuracy and precision	7	7	4	196	Use the b* values to assess steady-state condition. The value should stabilise and reach a plateau.	7	4	1	28
7	Data management	Manual data logging. The operator logs the data for each step change of the process to connect time point with process condition	Accuracy and precision	7	4	4	112	Implement standard operating procedures and automated data logging.	7	1	1	7
8	Data management	Data transfer and data integrity	Accuracy and precision	4	4	4	64	Save and copy the data for further analysis. Develop protocols that can be followed by operators.	4	1	1	4
9	Data analysis	Data not in steady state. Method validation outside limits of ATP (RMSE, R^2^, relative bias, repeatability, intermediate precision).	Accuracy and precision	10	4	7	280	Sample selection by applying PCA to the pre-filtered data from the experiment to verify if the steady state was reached.	10	4	1	40
10	Data analysis	Variabilities between samples. Method validation outside limits of ATP (Intermediate precision).	Accuracy and precision	10	7	4	280	Data normalisation by collecting polymer spectrum and use it as reference to normalise the sample spectra. This minimises variability between different day experiments.	10	4	1	40
11	Data analysis	Low signal to noise ratio. Method validation outside limits of ATP (RMSE, R^2^, relative bias, repeatability, intermediate precision).	Accuracy and precision	10	7	7	490	Spectral variable selection by identifying the parts of the spectra that are connected to change in amount of API using the loadings of PC1.	10	4	1	40
12	Data analysis	Under and over-fitting of PLS model depending on the number of latent variables used.	Accuracy and precision	10	7	7	490	Optimizing the number of latent variables to use by doing holdout cross-validation with 20% of the data set used to test and calculate RMSECV, R_cv_^2^.	10	1	1	10

S = severity, O = occurrence, D = detectability, RPN = risk priority number. Risk code of RPN by colour: red = high, yellow = medium, green = low, PCA = principal component analysis, API = active pharmaceutical ingredient, PC = principal component, PLS = partial least squares.

**Table 3 pharmaceutics-12-00150-t003:** Predicted PRX content in each validation day.

Day	True PRX Concentration (% *w*/*w*)	Mean Predicted PRX Concentration (% *w*/*w*)
1	11.56	11.59
2	11.75	11.70
1	13.46	13.44
2	13.44	13.59
1	15.44	15.31
2	15.46	15.50
1	17.50	17.38
2	17.49	17.57

**Table 4 pharmaceutics-12-00150-t004:** Trueness, precision and accuracy results for each concentration level in the validation data sets.

True PRX Concentration (% *w*/*w*)	Mean Predicted PRX Concentration (% *w*/*w*)	Trueness		Precision		Accuracy
		Relative Bias (%)	Recovery (%)	Repeatability (RSD%)	Intermediate Precision (RSD%)	Relative β-Expectation Tolerance Limits (%)
11.66	11.65	−0.12	99.88	0.55	0.84	[−2.92; 2.68]
13.45	13.51	0.47	100.47	0.54	0.95	[−3.51; 4.45]
15.45	15.40	−0.29	99.71	0.65	1.05	[−4.05; 3.47]
17.50	17.48	−0.14	99.86	0.80	1.09	[−3.13; 2.85]

**Table 5 pharmaceutics-12-00150-t005:** Error and uncertainty results for each concentration level in the validation data sets.

True PRX Concentration (% *w*/*w*)	Mean Predicted PRX Concentration (% *w*/*w*)	Error			Uncertainty		
		Absolute Total Error	Relative Total Error (%)	Uncertainty of the Bias (% *w*/*w*)		Measurement Uncertainty u(Y) (% *w*/*w*)	Expanded Uncertainty U(Y) (% *w*/*w*)
11.66	11.65	0.11	0.96	0.05		0.11	0.22
13.45	13.51	0.19	1.42	0.07		0.15	0.30
15.45	15.40	0.21	1.34	0.09		0.19	0.37
17.50	17.48	0.22	1.23	0.09		0.21	0.43

**Table 6 pharmaceutics-12-00150-t006:** Predicted PRX content in each DoE run.

True PRX Concentration (% *w*/*w*)	Feed Rate (g/min)	Screw Speed (rpm)	Mean Predicted PRX Concentration (% *w*/*w*)	Relative Bias (%)
14.45	5	150	13.77	−4.72
14.45	5	200	14.54	0.61
14.45	5	250	14.82	2.56
14.45	7	150	13.90	−3.77
14.45	7	200	14.60	1.04
14.45	7	200	14.60	1.01
14.45	7	250	14.94	3.37
14.45	9	150	13.28	−8.09
14.45	9	200	14.16	−2.00
14.45	9	250	14.59	0.95

**Table 7 pharmaceutics-12-00150-t007:** Method operable design region for feed rate and screw speed.

Feed Rate (g/min)	Screw Speed (rpm)	
	Lower Limit	Upper Limit
6	190	215
8	200	225

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
