# Peer review of "Development and Validation of an in-line API Quantification Method Using AQbD Principles Based on UV-Vis Spectroscopy to Monitor and Optimise Continuous Hot Melt Extrusion Process"

_pharmaceutics, 2020, doi:10.3390/pharmaceutics12020150_

Round 1

Reviewer 1 Report

Limitations of technology should be described. It only works for drug with high loading. what about drug required low drug loading? usually HME required plasticizers like TPGS or tween 80. such surfactants also absorbs UV. fig. 6 why absorbance is similar irrespective of drug loading? do absorbance of drug varies with presence of moisture, other component and temperature?

Author Response

Thank you for your valuable comments. We hope that we have addressed satisfactorily all the points that you have highlighted below.

Limitations of technology should be described. It only works for drug with high loading. what about drug required low drug loading?

Response: This technique works for a wide range of concentration including low ranges (results to be published soon). The high concentration values depend on the solubility of the API in the matrix used (lines 166-174) Advantages and limitations have been described in a previous publication (line 97).

Usually HME required plasticizers like TPGS or tween 80. such surfactants also absorbs UV.

fig. 6 why absorbance is similar irrespective of drug loading? do absorbance of drug varies with presence of moisture, other component and temperature?

Response: Absorbance is similar, but there are noticeable differences in the absorbance which enables quantification of the PRX across the ranges’ studies

Please see the response to the editors with the list of changes.

Reviewer 2 Report

The manuscript is well written with the appropriate flow. The present work demonstrates simple and reliable UV-Vis spectroscopy for HME. Data collection strategy, analysis, and robustness of methodology are prime importance for the continuous manufacturing formulation in industry. The work/protocol presented here is a good guideline for the industrial process and development compare to academic researcher. 

Author Response

Response: We are delighted that the reviewer found this paper worth to be published and it is a good guideline for industry and academia

Please see the response to the editors and other reviewers with the list of changes.

Reviewer 3 Report

This paper presents the development and validation of a quantitative method to predict the drug concentration during hot-melt extrusion using an in-line analytical quality tool.

The experimental presented in this paper is thorough and coherent, based on AQbD principles in accordance with the latest health standards. However, the validation showed in this paper sounds more technical than scientific and does not support the novelty required for a publication in a high-impact journal.

Moreover, the use of UV-VIS spectroscopy as a PAT in HME pharmaceutical production is relatively new, however, it has already been explored with the same drug product in the publication [12] in Pharmaceutics. Therefore, I cannot recommend publishing the article.

Author Response

Response: - Thank you for your comments. We would like to ask you to consider the points below.

AQbD concepts have not typically been applied to PAT methods and the use of the Analytical Target Profile to define and validate the performance requirements of an in-line method is novel. The approach to risk assess and improve control through RPN assessment whilst not novel has not been exemplified on a PAT method much (if at all) in the literature
- Publication [15, revised version] (from the same group) reports the use of the PAT method as a potential in-line system for early phase product development during pharmaceutical continuous manufacturing. Publication [15] also heavily focuses on the use of the method to optimise the process conditions. This new publication instead focuses on how to validate such method for routine use as a control method to monitor the performance of an optimised HME process. This is quite different to just using a method as a non-validated supporting tool in early development. Furthermore, the validation of such a method is not defined in ICHQ2(R1) which focuses on traditional batch analysis methods.  We believe the stepwise approach we took from defining the performance requirements, optimising the method and validating the predictive model would offer analytical scientists a useful case study on how to tackle the validation of similar methods deploying the use of PLS.

We have updated the manuscript to take into account the suggestions and comments from the editors and the other 3 reviewers. We believe this has improved the manuscript.

Please see the responses to the editors with the list of changes.

Reviewer 4 Report

This is an interesting paper that investigate the use of in-line UV analysis in HME.  Overall, I find the paper very difficult to follow and due to the lack of discussion I struggle to see any take home message or what this adds to the science.  I don't see the novelty in the paper as in-line analysis in HME using UV and Raman has been done before.  Based on this I have a number of questions and comments below.

Why did the authors use the drug piroxicam?  Can a statement confirming this be added to the paper.

A big issue with HME in thermal degradation.  Can this UV analysis be used to dectect and monitor impurity levels in the extrudate?

Introduction

Can you provide references for the statement that HME is used to enhance the solubility of APIs.

Can you add other references to the second last paragraph of the introduction that begins with Previous work.  Other groups such as Wesholowski et al and Schlindwein et al and already investigated this.

Materials and Methods

These are fine

Results and Discussion

Sections 3.1 to 3.1.3 read like methods rather than results and there is very limted discussion.  These should be moved to the methods section.

The results and discussion section is very difficult to follow and due to a lack of actual discussion I find it very difficult to understand the results.

Conclusion

The conclusion is weak and needs to be imporved.

Author Response

Thank you for your comments. Please find below the answers to the points that you have made:

Why did the authors use the drug piroxicam?  Can a statement confirming this be added to the paper.

Response: Thank you for your comments. Piroxicam is a BCS II drug (high permeability and low solubility) ideal model for product development of amorphous solid dispersions using HME process. More detail of API and polymer selection can be found in previous publications from the same research group (reference [15, revised version] and Lundsberg, L., et al. Process Analytical Technology (PAT), Chapter 9 in “Pharmaceutical Quality by Design: A Practical Approach” (editors, W Schlindwein and M Gibson), Wiley 2018).

A big issue with HME in thermal degradation.  Can this UV analysis be used to dectect and monitor impurity levels in the extrudate?

Response: We agree that thermal degradation is a very important issue in HME processes. This has been considered in previous publication [15 revised version]. Ref [15] reports the use of the PAT method as a potential in-line system for early phase product development during pharmaceutical continuous manufacturing. The paper heavily focuses on the use of the method to optimise the process conditions, i.e. prevent degradation products. This new publication instead focuses on how to validate such as method for routine use as a control method to monitor the performance of an optimised HME process.

Introduction

Can you provide references for the statement that HME is used to enhance the solubility of APIs.

Response: Yes, please see line 45

Can you add other references to the second last paragraph of the introduction that begins with Previous work.  Other groups such as Wesholowski et al and Schlindwein et al and already investigated this.

Response: Indeed, ref [15] is from our group. Refs [16-17] are from Wesholowski and they have been acknowledged. The approach and examples used in the latter publication are not the same as in this paper.

Materials and Methods

These are fine

Results and Discussion

Sections 3.1 to 3.1.3 read like methods rather than results and there is very limted discussion.  These should be moved to the methods section.

Response: We believe sections 3.1.1 - 3.1.2 are still better suited in the results section, but we do agree that we could add more discussion (see revised manuscript). We agree that section 3.1.3 should be moved to the Materials and Methods. Please see the changes marked in the manuscript. (line 165)

The results and discussion section is very difficult to follow and due to a lack of actual discussion I find it very difficult to understand the results.

Response: Please see the changes highlighted in the manuscript.

Conclusion

The conclusion is weak and needs to be imporved.

Response: Please see the changes highlighted in the manuscript.

Round 2

Reviewer 3 Report

Important experimental and mandatory aspects regard the validation of the analytical method are not properly explained. Moreover, seems that some important parameters are missing or erroneously performed according to ICH Q2 R2 as well as to FDA or other guidelines,

L165-180: A better explanation of how the analytical method validation was conducted should be added to the 2.6 section. Explain briefly how each validation parameter was performed (concentrations,  replicates, etc.). Some of that methodological information appeared inappropriately in the results and discussion section.

L391-401: Based on guidelines, intermediate precision should include different analysts. This evaluation was performed. Moreover, RSD should be calculated considering all variations on that level (days and analysts).   

L402-407: In linearity, additional statistical tests are required to evaluate the homogeneity of variance and the normality of residues. The figure 12 is completely unnecessary and should be removed.

L412: How the limit of quantification was determined? 

L443-478: Based on ICH Q2 R2, Robustness is “a measure of the method capacity to remain unaffected by SMALL, but deliberate variations in method parameters”. Thus, for this parameter, experimental should simulate, for example, the possible variations that the HME can assume under normal operating conditions. However, the variations used in the article, i.e. feed rate (from 5 to 9) and screw speed (from 150-250) are in a huge range. This protocol does not fit a robustness assay.

Specificity is a mandatory validation parameter. Please include this assay in the validation.

As a suggestion, two references that exemplify the application of the guidelines in a practical way and that could be used to support the discussion of the article are below.

Angelo, Tamara, et al. "Development and validation of a selective HPLC-UV method for thymol determination in skin permeation experiments." Journal of Chromatography B 1022 (2016): 81-86.

Pinho, Ludmila AG, et al. "Simultaneous determination of benznidazole and itraconazole using spectrophotometry applied to the analysis of mixture: a tool for quality control in the development of formulations." Spectrochimica Acta Part A: Molecular and Biomolecular Spectroscopy 159 (2016): 48-52.

Author Response

Comments and Suggestions for Authors

Important experimental and mandatory aspects regard the validation of the analytical method are not properly explained. Moreover, seems that some important parameters are missing or erroneously performed according to ICH Q2 R2 as well as to FDA or other guidelines,

 Thank you for your comments. Please see our responses below.

L165-180: A better explanation of how the analytical method validation was conducted should be added to the 2.6 section. Explain briefly how each validation parameter was performed (concentrations,  replicates, etc.). Some of that methodological information appeared inappropriately in the results and discussion section.

After round 1 of reviewers’ comments, section 2.6 was moved from ‘results’ to ‘materials and methods’ which was suggested by one of the reviewers. The explanation of the validation parameters is included in this section. Also, in the results part we have added table 3 with all the validation experiment results for each day.

L391-401: Based on guidelines, intermediate precision should include different analysts. This evaluation was performed. Moreover, RSD should be calculated considering all variations on that level (days and analysts).   

Analyst is mentioned in ICHQ2(R1) as a typical variation but it is not clear whether this is always needed. In a paper that we reference (ref 47) a risk-based approach for the design of intermediate precision studies is proposed which may or may not lead to designs that include analyst as a factor. We applied this risk-based approach which did not highlight analyst as a factor which required studying (due to the automated nature of the analytical procedure).

L402-407: In linearity, additional statistical tests are required to evaluate the homogeneity of variance and the normality of residues. The figure 12 is completely unnecessary and should be removed.

We understand your point, however, in this case additional statistical tests are not required because we are using high drug load (11.66 to 17.5). And this linear relationship is only valid in the validated range used.

We would like to keep Figure 12, as we believe it shows the relationship between predicted and true PRX concentrations in a clear way.

L412: How the limit of quantification was determined? 

As described in the ICHQ2R1, the quantitation limit is generally determined by the analysis of samples with known concentrations of analyte and by establishing the minimum level at which the analyte can be quantified with acceptable accuracy and precision. This paper does not include low drug content measurement. The method applied here is for high drug load, so the limit of quantification is the lowest value validated (11.66% w/w). The quantification limit is not required for an assay of content /potency (ref Table ICHQ2R1).

L443-478: Based on ICH Q2 R2, Robustness is “a measure of the method capacity to remain unaffected by SMALL, but deliberate variations in method parameters”. Thus, for this parameter, experimental should simulate, for example, the possible variations that the HME can assume under normal operating conditions. However, the variations used in the article, i.e. feed rate (from 5 to 9) and screw speed (from 150-250) are in a huge range. This protocol does not fit a robustness assay.

It is valid to determine robustness over a wider range if deemed useful. This approach is aligned with the MODR concept (which is currently being defined by the ICH expert working group in the evolving ICHQ14). We have included the text in line 475-476:

The robustness DoE was deliberately defined over a wider range (where the method performance meets the ATP [Error! Reference source not found.]) to gain further knowledge of the conditions and greater operational flexibility.

As a suggestion, two references that exemplify the application of the guidelines in a practical way and that could be used to support the discussion of the article are below.

Angelo, Tamara, et al. "Development and validation of a selective HPLC-UV method for thymol determination in skin permeation experiments." Journal of Chromatography B 1022 (2016): 81-86.

Pinho, Ludmila AG, et al. "Simultaneous determination of benznidazole and itraconazole using spectrophotometry applied to the analysis of mixture: a tool for quality control in the development of formulations." Spectrochimica Acta Part A: Molecular and Biomolecular Spectroscopy 159 (2016): 48-52.

Thank you. We have added both references. Line 91.

Submission Date

13 December 2019

Date of this review

27 Jan 2020 15:37:24

Reviewer 4 Report

Dear Authors,

Thank you for addressing my comments.  The paper is of suitbale standard for publication. However, I am still not convienced that the research is novel and of great interest.  I still feel that it's overall merit is low.

Author Response

Thank you very much.